# Zebrafish Embryonic Exposure to BPAP and Its Relatively Weak Thyroid Hormone-Disrupting Effects

**DOI:** 10.3390/toxics8040103

**Published:** 2020-11-13

**Authors:** Sangwoo Lee, Kojo Eghan, Jieon Lee, Donggon Yoo, Seokjoo Yoon, Woo-Keun Kim

**Affiliations:** 1Biosystem Research Group, Korea Institute of Toxicology, Daejeon 34114, Korea; sangwoo.lee@kitox.re.kr (S.L.); keghan@kitox.re.kr (K.E.); jieon.lee@kitox.re.kr (J.L.); donggon.yoo@kitox.re.kr (D.Y.); 2Human and Environmental Toxicology, University of Science and Technology, Daejeon 34113, Korea; sjyoon@kitox.re.kr; 3Department of Predictive Toxicology, Korea Institute of Toxicology, Daejeon 34114, Korea

**Keywords:** alternative chemical, BPA, endocrine disruption, thyroid hormone, zebrafish

## Abstract

Safe endocrine-disrupting alternatives for bisphenol A (BPA) are needed because its adverse health effects have become a public concern. Some bisphenol analogues (bisphenol F and S) have been applied, but their endocrine-disrupting potential is either not negligible or weaker than that of BPA. However, the endocrine-disrupting potential of bisphenol AP (BPAP), another BPA alternative, has not yet been fully assessed. Hence, we evaluated the thyroid hormone (TH)-disrupting potency of BPAP because THs are essential endocrine hormones. Zebrafish embryos were exposed to BPAP (0, 18.2, 43.4, or 105.9 μg/L) for 120 h, and TH levels, the transcription of 16 TH-related genes, the transcriptome, development, and behavior were evaluated. In our study, a decrease in T4 level was observed only at the maximum nonlethal concentration, but significant changes in the T3 and TSHβ levels were not detected. BPAP did not cause significant changes in transcription and gene ontology enrichment related to the TH system. Developmental and behavioral changes were not observed. Despite T4 level reduction, other markers were not significantly affected by BPAP. These might indicate that BPAP has weak or negligible potency regarding TH disruption as a BPA alternative. This study might provide novel information on the TH-disrupting potential of BPAP.

## 1. Introduction

The endocrine-disrupting effects of bisphenol A (BPA) and its major alternatives, such as bisphenol F (BPF) and bisphenol S (BPS), have recently been reported [1,2,3]. As one of the integral endocrine systems, the thyroid hormone (TH)-disrupting effects of those chemicals need to be evaluated because THs play essential roles in many physiological processes, such as development, growth, reproduction, and metabolism [3,4]. BPA and other bisphenols, such as bisphenol AF (BPAF), BPF, and BPS, caused the disruption of the TH regulation system [1,2,3,4,5]. In our previous study, BPA, BPF, and BPS increased the level of T3 or T4 and changed related gene transcriptions in the zebrafish embryo/larvae model [3]. BPA, BPF, and BPS also induced the proliferation of TH-dependent rat pituitary cells (GH3) and biphasic responses of gene transcription in *Pelophylax nigromaculatus* tadpoles [2]. However, T4 level reduction was also shown by BPF and BPS [1,4,5]. Despite conflicting results for bisphenols regarding THs, the TH-disrupting effects of those bisphenols are evident. To find alternative chemicals that are relatively safer than BPA with respect to TH disruption, various candidates need to be evaluated.

As an alternative candidate, bisphenol AP (BPAP), which is also known as 4,4′-(1-phenylethylidene) bisphenol, is used in polymer materials, the fine chemical industry, and the medicine industry. As an indispensable plasticizer and flame retardant, BPAP is used in the synthesis of plastic, rubber, and other industrial products [6]. The widespread use of these synthetic products has led to BPAP contamination of environments and foods, and therefore, the exposure of humans to this chemical [7]. BPAP was recently detected in various samples of personal care products and foods in China and the United States [8,9]. In a previous study, BPAP was detected in 10% of 231 personal care products, and based on the detection rate, BPAP was ranked third among eight bisphenols, following BPA and BPS [8]. In 289 food samples, BPAP was ranked third among bisphenols based on its mean concentration, following BPA and BPF, and the mean concentration of BPAP was relatively high in eggs and dairy products [9]. BPAP was found in human urine samples from Saudi Arabia at a range of 0.063 to 12.6 ng/mL [10] and from China at a range of <LOQ to 2.05 ng/mL [11,12]. In addition, BPAP was detected at up to 56 ng/L and 1.2 ng/g dw in surface water from Luomo lake and sediment samples from Taihu lake, respectively [13]. In Korean domestic wastewater treatment plants, the measured concentration of BPAP was up to 16.0 ng/g dw [14]. However, the environmental levels and detection frequencies of BPAP were not higher than BPA and major alternatives of BPA, such as BPF and BPS [13,14].

The toxicological information on BPAP remains insufficient. Regarding its TH-disrupting effects, only two in vitro studies have been performed [15,16], and the effects are unclear. These previous studies revealed that BPAP at levels up to the maximum non-cytotoxic dose did not affect gene transcription in rat thyroid follicular cells (FRTL-5) [15], and the proliferation of GH3 cells was not changed under the T3 cotreatment conditions [16]. In contrast, under conditions without T3, treatment with the maximum non-cytotoxic dose of BPAP increased GH3 cell proliferation and decreased gene transcription (*trα*, *trβ*, and *dio2*) [15,16].

Because the TH-disrupting effects of BPAP remain unclear, an evaluation of TH disruption using more comprehensive in vivo models, e.g., zebrafish, is needed due to these models having a complete TH regulation system, such as hypothalamus–pituitary–thyroid (H–P–T) feedback. As a non-animal alternative model for chemical screening, zebrafish embryos can be used until the zebrafish starts to feed independently, which is at 5 d post fertilization (dpf) [17]. Despite morphological differences in its mature thyroid gland compared to higher vertebrates, the early steps in thyroid development of zebrafish show a significant resemblance [18,19]. From 70 to 80 h post fertilization (hpf), T4 synthesis in zebrafish embryo becomes detectable [19,20].

The aim of this study was to evaluate the integrated TH-disrupting effects of BPAP, including phenotypic endpoints (development and behavior). Based on a previous review, an integrated approach that included a combination of morphological, behavioral, and molecular process assessments appears to be the most suitable for facilitating an understanding of the TH-disrupting effects [19]. In addition, we compared the TH-disrupting potency of BPAP with that of BPA and other alternatives to determine whether BPAP is relatively safe.

## 2. Materials and Methods 

### 2.1. Test Chemicals and Chemical Analysis

BPAP (CAS RN. 1571-75-1) was purchased from Sigma Aldrich (St. Louis, MO, USA) (Appendix A) and was dissolved in dimethyl sulfoxide (DMSO) at 0.1% *v*/*v*. The exposure media were collected before and after the media were changed. It was vortexed weakly and separated by centrifuging for 5 min at 12,000 rpm. Then, the samples were filtered through 0.22 μm filters for quantification. 

The actual concentrations of BPAP in the exposure media were measured by high-performance liquid chromatography (HPLC; Agilent 1260 Infinity, Agilent Technologies, Palo Alto, CA, USA) coupled with an autosampler and diode array detector. The separation was performed using a Waters SunFire^TM^ C18 column (100 × 4.6 mm, 5 μm) with an isocratic mobile phase consisting of 30% water and 70% methanol. The flow rate was set to 1 mL/min. The autosampler and column oven temperatures were maintained at 4 and 50 °C, respectively. The detection wavelength was set to 275 nm, and the final injection volume of all the samples was 25 μL. The calibration curve for quantifying BPAP was linear, in the range of 10–3000 ng/mL, and the limit of quantification (LOQ) was determined as LOQ = 10 × (SD/Slope), where SD is the standard deviation of the response, and slope is slope of the calibration curve. The measured concentrations are shown in Appendix A and were used to present the results obtained in our study.

### 2.2. Zebrafish Culture and Exposure Design

Zebrafish (*Danio rerio*) embryos were obtained from mating pairs of wild-type adult zebrafish that were cultured under a constant temperature (28 ± 1 °C) and a 14-h light/10-h dark photoperiod by the Biosystem Research Group at the Korea Institute of Toxicology (Daejeon, Korea). The embryos were collected and inspected under a stereomicroscope to select normally fertilized and developed embryos. The test solutions were prepared with E3 media, which contained 0.292 g of NaCl, 0.013 g of KCl, 0.044 g of CaCl_2_·2H_2_O, and 0.081 g of MgSO_4_·7H_2_O in Millipore-filtered water (1 L). The exposure concentrations, i.e., solvent control (SC) and 18.2, 43.4, and 105.9 μg/L BPAP, were determined as nonlethal concentrations based on the preliminary test (Appendix A). In our preliminary test, significant lethal effects were observed at 398.9 µg/L of BPAP but not at 105.9 µg/L of BPAP (>95% survival rate was observed).

Embryos (≤3 hpf) were randomly distributed and exposed to the test chemical or the SC until 120 hpf, and the exposure medium was renewed every other day during the exposure period. For observations of hatching, growth, and malformation, the embryos were placed in 96-well plates. Triplicates of each treatment were prepared, and each replicate included eight larvae (one larva/well). To measure the TH level in zebrafish larvae, embryos (200 larvae per replicate, four replicates per treatment) were placed in a glass beaker (volume of 250 mL). Another set of exposures using a glass beaker (volume of 50 mL) was performed using triplicate samples (25 larvae per replicate) per treatment for transcription and transcriptome analysis, respectively. All exposure experiments were conducted in agreement with protocols approved by the Institutional Animal Care and Use Committee (IACUC) of the Korea Institute of Toxicology (Protocol No. KIT-1907-0263, approval date 29 July 2019). This experiment also complied with the ARRIVE guidelines [21] and was performed in accordance with the U.K. Animals (Scientific Procedures) Act of 1986 and associated guidelines, and the EU Directive 2010/63/EU for animal experiments [17].

### 2.3. TH Measurement

Following the exposure of embryos/larvae at 120 hpf, TH levels were measured from the homogenates of pooled fish (*n* = 200/replicate) using an enzyme-linked immunosorbent assay (ELISA) according to the protocol described by Yu et al. (2010) [22] with minor modifications. Briefly, 200 zebrafish larvae in a glass beaker were homogenized in 200 µL of 1× PBS using a pestle tissue grinder. The samples were then centrifuged for 10 min at 5000 × *g* and 4 °C. The supernatant of the samples was collected and stored at −80 °C until further measurement. The levels of hormones were quantified using ELISA kits (Cusabio Biotech, Wuhan, China) for T4 (Cat no. CSB-E08489f), T3 (Cat no. CSB-E08488f), and TSHβ (Cat no. CSB-EQ027261FI) with a BioTek Cytation 5 system (BioTek, Winooski, VT, USA) according to the manufacturer’s recommended protocol. The optical density for T4, T3, and TSHβ was determined using BioTek Cytation 5 system (BioTek, Winooski, VT, USA) set to 450 nm. The limits of detection for T4, T3, and TSHβ are reportedly 20 ng/mL, 0.5 ng/mL, and 2.5 µIU/mL, respectively. The measurements of T4, T3, and TSHβ were normalized to the protein level (mg/mL). Protein levels were analyzed using a BCA protein assay kit (Thermo Fisher Scientific, Johannesburg, South Africa) and BioTek citation 5 (BioTek, Winooski, VT, USA) system set to 562 nm.

### 2.4. RNA Isolation and Quantitative RT-PCR

For quantitative RT-PCR, 25 zebrafish larvae exposed to each replicate treatment were pooled and homogenized using a pestle tissue grinder. Total RNAs of the samples were then extracted using an RNeasy mini kit (Qiagen, Hilden, Germany). The total RNA concentration of each sample was detected using an ND-1000 spectrometer (NanoDrop Technologies, Wilmington, WD, USA). Reverse-transcribed complementary DNAs (cDNAs) were prepared using the iScript^TM^ cDNA synthesis kit (BioRad, Hercules, CA, USA) and then, diluted to 300 ng/µL. Quantitative real-time PCR (qRT-PCR) was performed using a StepOnePlus real-time PCR system (Applied Biosystems, Foster City, CA, USA) with a qRT-PCR mix (a total of 20 µL) that included an ABI SYBR Green Master Mix (10 µL) (Applied Biosystems, Foster City, CA, USA), 10 pmol of the PCR primers (1.8 µL of each), purified PCR-grade water (4.4 µL), and the cDNA sample (2 µL). The thermal cycle profile was preincubation at 95 °C for 10 min followed by 40 cycles of amplification at 95 °C for 10 s, 60 °C for 20 s, and 72 °C for 20 s. A melting curve analysis was performed to confirm the melting temperatures of the PCR products. The comparative Ct method (2^−ΔΔCt^) was applied to calculate the relative transcription levels [23]. Sixteen genes related to thyroid stimulation (*crh* and *tshβ*), thyroid development (*nkx2.1*, *hhex*, *tshr*, *slc5a5*, *tg*, *pax8*, and *tpo*), TH receptors (*trα* and *trβ*), TH transport (*ttr*), and TH metabolism (*dio1*, *dio2*, *dio3*, and *ugt1ab*) were selected. Due to its stable expression during the development of zebrafish, *18S rRNA* was selected as the reference gene [24]. The primer sequences for the reference and target genes are shown in Appendix A.

### 2.5. RNA-Sequencing and Ingenuity Pathway Analysis (IPA)

To elucidate the pathways underlying the induction of toxicity in response to BPAP exposure, an IPA was performed after RNA sequencing. The IPA results were obtained from a comparison between the SC group and the maximum nonlethal concentration group (105.9 µg/L BPAP). RNA sequencing was conducted according to a previous study [25], and more detailed information is presented in the Appendix A. An integrated analysis of toxicity functions and canonical pathways was conducted using IPA software (IPA, Qiagen, Hilden, Germany) (*p* < 0.1) [26].

### 2.6. Hatchability, Growth, and Morphological Observations

The survival, hatching, and malformation of embryos/larvae were observed and recorded every 24 h during the exposure period. The incidences of mortality, hatching, and malformation were calculated in triplicate per concentration. After exposure, the body lengths and eyeball sizes of the zebrafish larvae were also measured (*n* = 5). For the measurement, the larvae were anesthetized by tricaine methane sulfonate (0.004% *w*/*v*) and transferred to a glass slide containing methylcellulose. Then, individual images were taken using a Leica M205FA fluorescent microscope mounted with a Leica DFC 7000T camera module (Leica Camera AG, Wetzlar, Germany).

### 2.7. Behavior Analysis (Locomotor Activity)

Following BPAP exposure, the movement of zebrafish was monitored using an automated tracking device (DanioVision, Noldus Information Technology, Wageningen, The Netherlands). Four embryos/larvae were allocated to each replicate of each treatment group (SC and 18.2, 43.4, and 105.9 µg/L BPAP, *N* = 6), and after 6 min of acclimation, the tracking was continued for four phases, which consisted of two cycles of a 6-min light/6-min dark period for a total of 24 min. Using EthoVision software (Noldus Information Technology, Wageningen, The Netherlands), the distance moved (cm) and moving duration (s, time of moving speed > 0.2 cm/s) were calculated for each individual larvae [27].

### 2.8. Statistical Analysis

The normality of the distribution and the homogeneity of variance were analyzed using the Shapiro–Wilk test and Levene’s test, respectively. To identify significant differences between the treatments and the SC, one-way analysis of variance (ANOVA) with Dunnett’s and Dunnett’s T3 post hoc analysis was performed using SPSS 12.0 for Windows^®^ (SPSS, Chicago, IL, USA) for equal variances and unequal variances, respectively. The analysis of significant differences in nonparametric data was performed using the Kruskal-Wallis test. Differences with *p* < 0.05 were considered statistically significant.

### 2.9. Integrated Comparison with Other Bisphenols

The results obtained in the present study were compared with those found for other bisphenols (BPA, BPF, BPS, and BPZ) in a previous study [3]. The maximum fold changes of multiple endpoints regarding TH disruption, e.g., gene transcription and hormonal levels, are presented using a star plot. A score of the integrated endpoints was calculated from the area of the star plot, i.e., the summation of multiple triangular areas consisting of two clockwise consecutive endpoints from the center [28].

## 3. Results

### 3.1. Changes in the TH and TSHβ Levels

In our study, BPAP decreased the T4 level in a dose-dependent manner, and only the maximum nonlethal concentration (105.9 µg/L BPAP) induced a significant reduction (1.5-fold) (Figure 1a). However, significant effects on the T3 and TSHβ levels in the zebrafish larvae were not observed in the present study (Figure 1b,c).

### 3.2. Transcriptional Changes in TH-Related Genes

BPAP exposure did not change the transcription of 16 genes related to the thyroid regulation system. Specifically, the transcription of genes involved in thyroid stimulation, i.e., *crh* and *tshβ*, was not significantly altered (Table 1). Among the genes related to TH synthesis, receptors, and transport, the transcription of the *tg* and *trα* genes in the zebrafish exposed to BPAP was more than 1.5-fold higher than that in the SC-treated zebrafish, but this difference was not statistically significant (Table 1). In addition, no significant changes in TH metabolism-related genes were detected (Table 1).

### 3.3. RNA Sequencing and IPA

We found 32 toxicity functions with a −log(*p*-value) over 1.0 from the IPA followed by RNA sequencing. Among these functions, the highly affected pathways were found to be related to renal function, cardiac function, and liver function (Figure 2). Toxicity functions directly linked to TH disruption were not found. Pathways linked to TH system regulation were also not observed among the canonical pathways derived from BPAP-exposed zebrafish larvae (Appendix A).

### 3.4. Changes in Survival, Hatchability, Growth and Morphology

Significant lethal effects were observed after exposure to 398.9 µg/L BPAP. Exposure to BPAP at levels up to 105.9 µg/L exerted no significant effects (>95%) on the survival of zebrafish larvae (Appendix A). Thus, other observations (time to hatching, body length, eyeball size, morphology, and behavior) and analyses (TH analysis, gene transcription, and transcriptome) were performed after exposure to the SC and BPAP at concentrations up to 105.9 µg/L.

All surviving embryos were successfully hatched (data not shown). The time to hatching was not significantly affected by exposure to BPAP at levels up to the maximum nonlethal concentration (105.9 µg/L) (Figure 3a). Additionally, BPAP did not exert any effects on malformation (Appendix A), body length, or eyeball size (Figure 3b,c)).

### 3.5. Behavioral Changes

The locomotor activity of zebrafish larvae exposed to BPAP did not differ from that of the SC group (Figure 4 and Appendix A). Exposure to BPAP to 120 h did not significantly alter the distance moved (cm), although exposure to 105.9 µg/L BPAP decreased this distance in the dark phases by 14% compared with that obtained with the SC (Figure 4a). The moving duration (s, time of movement at a speed > 0.2 cm/s) was also not affected by BPAP exposure (Figure 4b).

### 3.6. Integrated Comparison with Other Bisphenols

The potency of the TH-disrupting effects of BPAP was relatively lower than that of the effects of other bisphenols (Figure 5), such as BPA, BPF, BPS, and BPZ [3]. The scores obtained from the integrated endpoints after BPS, BPF, BPA, BPZ, and BPAP exposure, i.e., the area of the star plot, were 11.5, 8.9, 7.3, 7.0, and 5.7, respectively. The comparison of these scores showed that none of the other bisphenols yielded a score lower than that obtained for BPAP.

## 4. Discussion

### 4.1. Effects of BPAP on the TH Levels

The physiological effects of BPAP on the TH system observed in our study were not substantial, although a significant reduction in T4 level (1.5-fold) was detected in zebrafish larvae exposed to the maximum concentration (105.9 µg/L) of BPAP. A change in the level of T4, which is the precursor of T3, can be regarded as a signal of TH disruption [29]. However, the physiological effects on the TH system should be assessed based on the T4, T3, and TSHβ levels rather than the T4 level alone [30]. In this study, the levels of TSHβ and T3 were not altered. T3, which is converted from T4 in peripheral tissue, is very important as the active form of TH [31]. TSHβ, the major regulator of TH synthesis, is secreted from the pituitary and acts as a stimulation signal for TH synthesis in the thyroid gland [4]. Based on an analysis of the levels of T4, T3, and TSHβ together, the effects of BPAP on the TH regulation system are not clearly substantial.

Additionally, circumambient components around T4 did not support the TH-disrupting potential of BPAP. The reduction in T4 can be mainly attributed to a decrease in TH synthesis or an increase in the metabolism of T4 [1,32,33]. Based on our results, the transcription of genes related to thyroid stimulation and TH synthesis was not significantly changed (Table 1). As mentioned above, the hormonal level of TSHβ was also not decreased by BPAP (Figure 1c). The excess metabolism of the T4 hormone could be assessed by detecting the transcription of deiodinase (*dio1*, *dio2*, and *dio3*) and UDP-glucuronosyltransferase (*ugt1ab*). The transcription of these genes was also not significantly affected (Table 1). Despite the reduction in the T4 level, the TH-disrupting potential of BPAP is not sufficiently clear.

### 4.2. Effects of BPAP on Gene Transcription and the Transcriptome

The effects of BPAP on TH disruption were not supported by the results from the analysis of gene transcription. In the zebrafish larvae exposed to BPAP, no significant changes in the transcription level of 16 genes related to thyroid stimulation and TH synthesis, receptors, transport, and metabolism were observed (Table 1). These genes were widely used as indicators of TH disruption in previous studies [1,3,4].

Among these genes, the transcription of the *tg* and *tr* genes was increased by more than 1.5-fold by exposure to BPAP. As a precursor of TH, thyroglobulin, i.e., *tg*, has been proposed as a thyroid abnormality [34]. The direction of *tg* transcription observed in our study might be interpreted as compensation for the reduction in T4 level because upregulation of the *tg* gene could promote thyroid development to compensate for reductions in the T4 levels [4]. Previous studies showed that BPAP increased the transcription of the *tg* gene and decreased the T4 level [1,31,32,33]. The direction of *tg* gene transcription observed in our study was in accordance with that found in previous studies. However, our result did not reach statistical significance. TH receptors (TH receptor α (trα) and TH receptor β (trβ)) are members of the superfamily of ligand-dependent transcription factors [35]. When T3 binds to the TH receptor dimer, it activates gene transcription by releasing corepressors and recruiting coactivators [36]. In our study, *trα* transcription was upregulated by more than 1.5-fold after exposure to BPAP, but this difference was not statistically significant. Previous studies showed that decabromodiphenyl ether (BDE-209) and arsenite significantly increased the transcription of *trα* and decreased the T4 level [34,37]. However, this finding might be due to an increase in T3, which was observed in response to decreases in the T4 level in both previous studies [34,37]. In the abovementioned previous studies, significant changes in deiodinase genes (*dio1* and *dio2*) that convert T4 to T3 were also observed [34,37]. Another previous study also indicated that the administration of T3 induced the transcription of *trα* and *trβ* [38]. In our study, BPAP did not affect the T3 level in zebrafish larvae.

The transcriptome analysis does not support the disruption of TH by BPAP. Specifically, BPAP did not induce changes in transcriptome-related TH disruption. RNA sequencing and IPA provide the ability to map differentially expressed proteins to fixed canonical pathways [26]. In addition, an analysis of toxicity functions by IPA reveals biological mechanisms that are related to toxicity [26]. We conducted RNA sequencing and IPA to observe the changes in the whole transcriptome between the SC group and the 105.9 μg/L BPAP exposure group. Pathways directly related to the TH system were not found among the obtained lists of toxicity functions and canonical pathways (*p*-value < 0.1) (Figure 2 and Appendix A). The major toxicity functions identified to be associated with BPAP exposure were mainly related to renal, heart, and liver damage, cardiac inflammation, or liver necrosis/cell death. Surely, THs are involved in various physiological functions. For example, hypothyroidism is accompanied by a decrease in glomerular filtration, hyponatremia, and alteration of water excretion ability [39,40]. Additionally, THs are known to modulate the components of the cardiovascular system necessary for normal cardiovascular development and function [41]. The disruption of cellular TH signaling triggers chronic liver diseases and nonalcoholic fatty liver disease [42,43]. However, because toxicity functions and canonical pathways related to the TH system were not identified from the transcriptome changes, the transcriptome changes induced by BPAP exposure were not unlikely changed by TH disruption. The overall gene transcription and RNA sequencing results obtained in our study might imply that the thyroid disruption obtained with BPAP is not sufficiently strong to induce the following toxicity functions.

### 4.3. Effects of BPAP on Hatching, Growth and Behavior

Even at the phenotypic level, the TH-disrupting effects of BPAP were not confirmed in our study. Phenotypic endpoints were suggested as one of the components in the integrated approach for the suitable assessment of TH disruption [19]. A previous study suggested hatching, body length, eye impairment, morphological effects, and behavioral effects as indicators that should be monitored with the endpoints of TH disruption and demonstrated that these indicators show responses of 82%, 58%, 42%, 87%, and 69%, respectively, to substances that cause TH disruption [19]). Bisphenol F, S, and Z, which have been used as BPA analogs similar to BPAP, also induced changes in the time to hatch or a reduced eyeball size ratio accompanied by a significant TH increase [3]. In the present study, however, zebrafish exposed to BPAP did not exhibit significant changes in hatching, body length, eyeball size, or morphological abnormalities (Figure 3 and Appendix A).

Behavior might also be affected by TH-disrupting substances, and the assumption that TH disruption is associated with neurotoxicity was previously proposed [44]. In addition, behavioral assessment has been used as an observation point more frequently in thyroid-related studies than in other type of studies [19]. In our study, BPAP failed to result in significant changes in the locomotor activity (distance moved) of zebrafish (Figure 4a). Consistent with the typical nature of zebrafish, increased movement in the dark phase compared with the light phase was found in the present study (Figure 4a). In addition, BPAP exposure did not affect the moving duration, which indicates that the zebrafish moved faster than 0.2 cm/s (Figure 4b). Based on the overall phenotypic indicators, i.e., hatching, body length, eye impairment, morphological effects, and behavioral effects, our results might indicate that the TH-disrupting effects of BPAP are negligible.

### 4.4. Disruption Potency of BPAP in Comparison with that of Other Bisphenols

The TH-disrupting effects of BPAP appeared to be relatively lower than those of other bisphenols (Figure 5). Although the weight of individual markers used to determine the score of integrated endpoints might not be equal, we believe that the relative comparison of integrated molecular levels might provide insight into TH disruption-safer chemicals, and our results might indicate that BPAP is relatively safer than other bisphenols with respect to TH disruption. Based on previous studies, even the estrogen receptor α binding and genotoxic potencies of BPAP were lower than those of BPA, although further in-depth study is needed [45,46].

## 5. Conclusions

Despite the observed reduction in the T4 level, other markers for gene transcription, transcriptome, development, and behavior were not significantly affected in zebrafish exposed to BPAP at levels up to sublethal concentrations. Thus, our results might indicate that BPAP exhibits negligible or weak potency for TH disruption. In addition, BPAP exerted a relatively lower impact on TH disruption than other bisphenols, i.e., BPA, BPF, BPS, and BPZ. Thus, based on its TH-disrupting effects, BPAP could be considered a safer alternative chemical to BPA. This study can provide novel information about the potential thyroid-disrupting effects of BPAP and might help society develop and select safer substitutes.

## Figures and Tables

**Figure 1 toxics-08-00103-f001:**
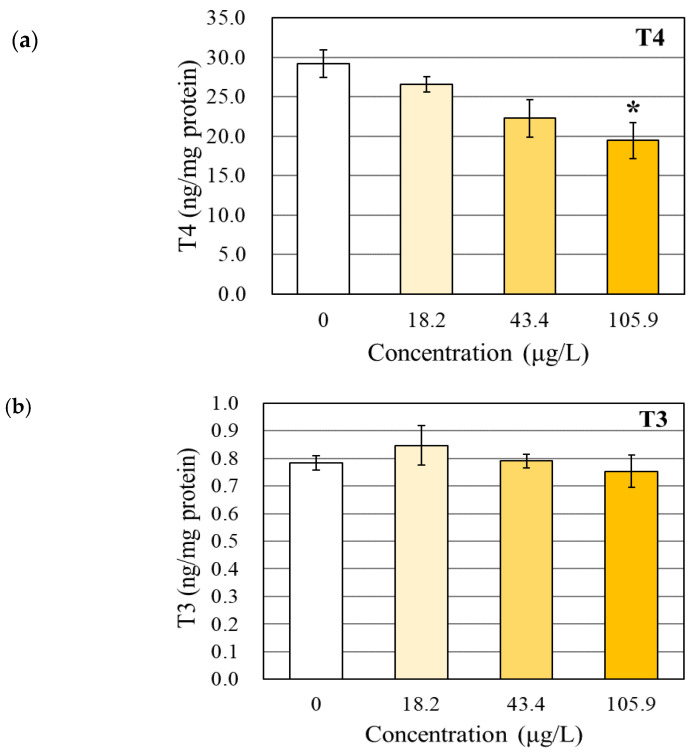
Hormonal levels in whole-body homogenates of zebrafish larvae after bisphenol AP (BPAP) exposure ((**a**) T4, (**b**) T3, and (**c**) TSHβ). The results are shown as the means ± SEMs (*N* = 4). Asterisks (*) indicate significant differences compared with the solvent control (0.1% *v*/*v* DMSO) (*p* < 0.05).

**Figure 2 toxics-08-00103-f002:**
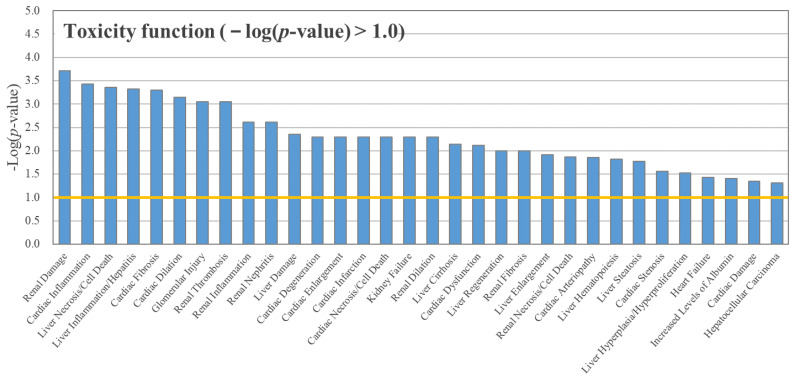
Toxicity functions in zebrafish larvae after BPAP exposure (105.9 μg/L). Toxicity functions with −log (*p*-value) values higher than 1.0 (*N* = 3) were used in the ingenuity pathway analysis (IPA).

**Figure 3 toxics-08-00103-f003:**
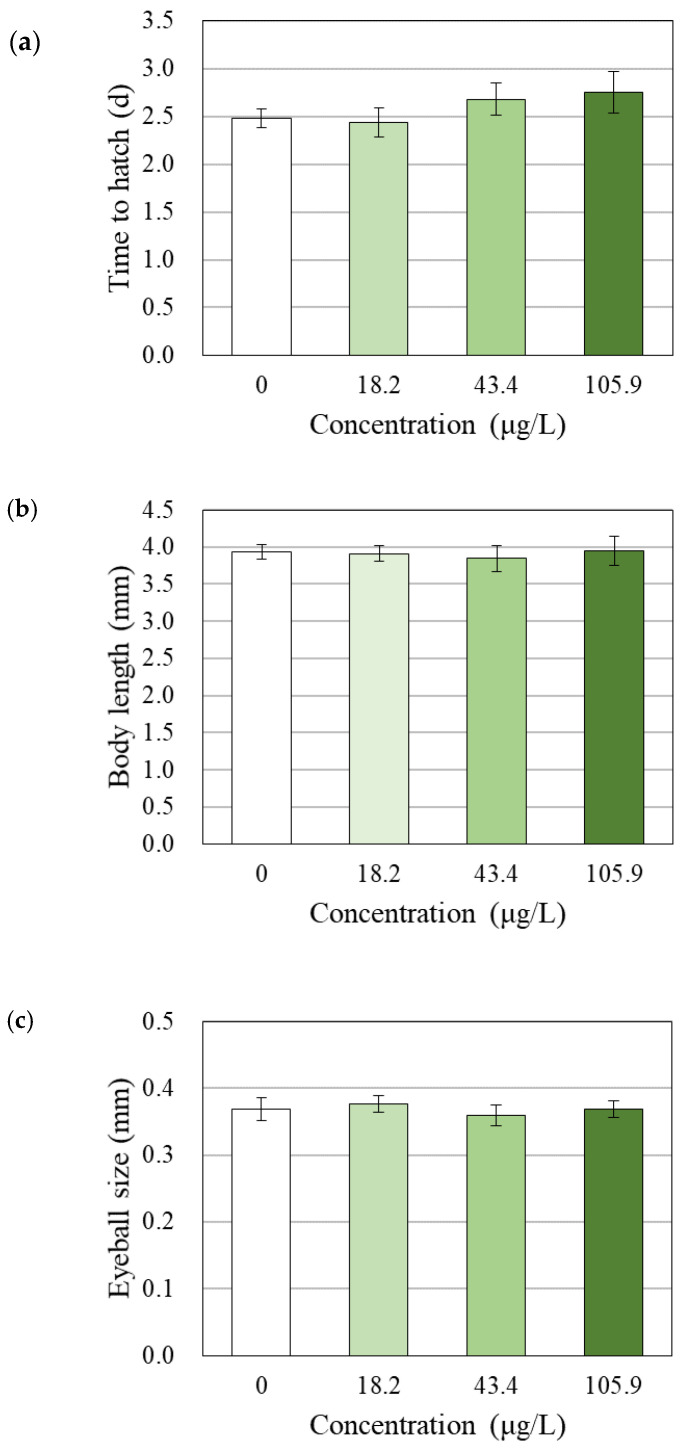
Effects of BPAP on (**a**) the time to hatch, (**b**) body length, and (**c**) eyeball size (*N* = 3 for time to hatch, *N* = 5 for body length and eyeball size). The times to hatch, body lengths, and eyeball sizes are shown as the means ± SDs. None of the effects were statistically significant compared with the solvent control (0.1% *v*/*v* DMSO) (*p* < 0.05).

**Figure 4 toxics-08-00103-f004:**
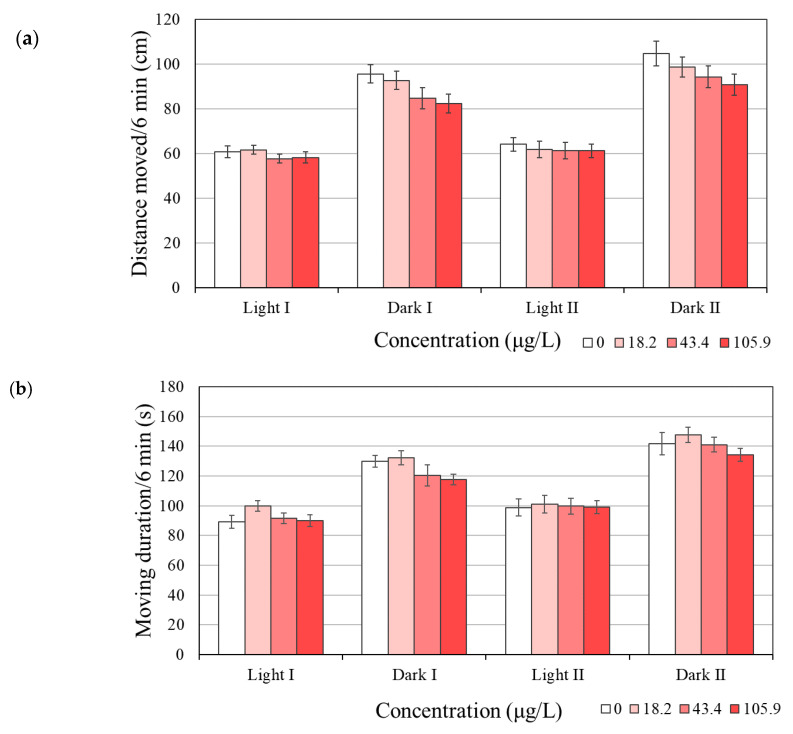
Locomotor activity ((**a**) distance moved and (**b**) moving duration) of zebrafish larvae after BPAP exposure. The moving duration was identified as the amount of time that the zebrafish moved at speeds over 0.2 cm/s. The results are shown as the means ± SEMs (*N* = 6). None of the effects were statistically significant compared with the solvent control (0.1% *v*/*v* DMSO) (*p* < 0.05).

**Figure 5 toxics-08-00103-f005:**
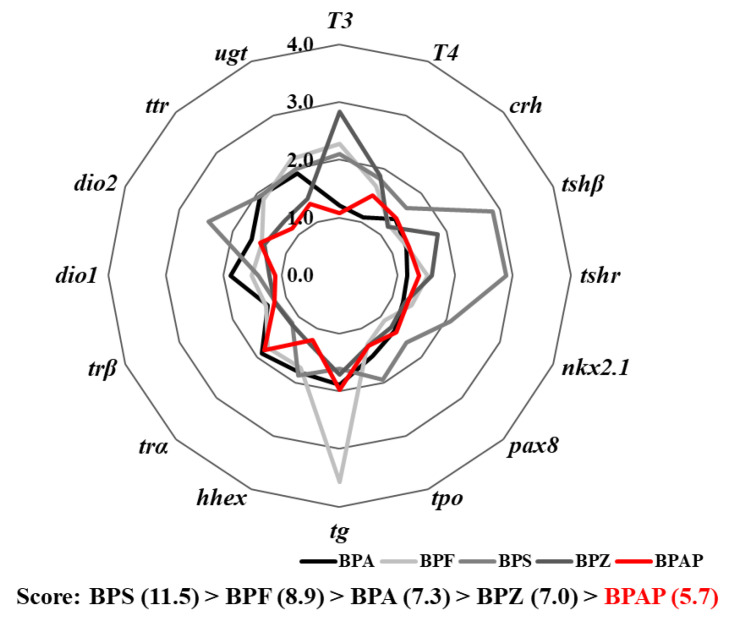
Comparison of the thyroid hormone-disrupting potential of BPAP with those of previously reported bisphenols, i.e., BPA, BPF, BPS, and BPZ (Lee et al., 2019).

**Table 1 toxics-08-00103-t001:** TH-related gene transcription levels (fold changes) in whole-body homogenates of zebrafish larvae after BPAP exposure ((a) thyroid stimulation, (b) TH synthesis, (c) TH receptors and transport, and (d) TH metabolism).

Conc. (μg/L)	Thyroid Stimulation		TH Receptors and Transport				
	*crh*	*tshβ*	*trα*	*trβ*	*ttr*		
0	1.00 ± 0.21	1.00 ± 0.23	1.00 ± 0.34	1.00 ± 0.08	1.00 ± 0.08		
18.2	0.72 ± 0.18	0.79 ± 0.27	1.43 ± 0.37	1.02 ± 0.24	1.00 ± 0.15		
43.4	0.89 ± 0.16	0.77 ± 0.26	1.83 ± 0.31	1.21 ± 0.20	1.15 ± 0.26		
105.9	1.13 ± 0.10	1.29 ± 0.20	1.57 ± 0.28	1.03 ± 0.25	1.16 ± 0.22		
**Conc.** **(μg/L)**	**TH Synthesis**						
	***nkx2.1***	***hhex***	***tshr***	***slc5a5***	***tg ^a^***	***Pax8***	***tpo***
0	1.00 ± 0.23	1.00 ± 0.15	1.00 ± 0.19	1.00 ± 0.17	1.02 (0.92–1.07)	1.00 ± 0.26	1.00 ± 0.08
18.2	0.79 ± 0.25	0.83 ± 0.16	0.72 ± 0.25	0.60 ± 0.23	0.86 (0.35–1.01)	0.72 ± 0.10	0.76 ± 0.06
43.4	0.86 ± 0.26	0.94 ± 0.13	0.77 ± 0.31	0.55 ± 0.23	1.03 (0.93–2.25)	0.92 ± 0.15	0.98 ± 0.07
105.9	1.26 ± 0.26	1.21 ± 0.27	1.19 ± 0.27	1.04 ± 0.27	1.60 (1.47–2.88)	0.78 ± 0.19	0.81 ± 0.11
**Conc.** **(μg/L)**	**TH Metabolism**						
	***dio1***	***dio2***	***dio3 ^a^***	***ugt1ab***			
0	1.00 ± 0.19	1.00 ± 0.25	0.99 (0.89–1.12)	1.00 ± 0.17			
18.2	0.97 ± 0.03	0.67 ± 0.22	0.79 (0.45–1.16)	0.75 ± 0.10			
43.4	1.11 ± 0.13	0.93 ± 0.01	0.88 (0.71–0.88)	0.91 ± 0.08			
105.9	1.11 ± 0.03	1.13 ± 0.28	1.03 (0.87–2.29)	1.17 ± 0.19			

The results are shown as the means ± SEMs (*N* = 3). None of the effects were statistically significant compared with the solvent control (0.1% *v*/*v* DMSO) (*p* < 0.05). ^*a*^: Nonparametric data shown as the median with ranges.

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
