# Peer review of "Zebrafish Embryonic Exposure to BPAP and Its Relatively Weak Thyroid Hormone-Disrupting Effects"

_toxics, 2020, doi:10.3390/toxics8040103_

Round 1

Reviewer 1 Report

This study is focused on the effects of bisphenol AP (BPAP), a native bisphenol analogue, in zebrafish development by evaluating developmental, behavioural, biochemical, and genetic parameters after exposure to different BPAP concentrations. The developmental and behavioural outcomes were not affected by the exposure and no changes were observed in the transcriptional profile of different genes although slight changes. The main effect observed was the increased T4 hormone levels which translated as a weak or negligible TH disruption of this compound in zebrafish.

Overall, the manuscript present novel and significant scientific data for the risk assessment of this alternative to bisphenol and, despite being a well-designed work presenting an interesting and cutting-edge approach, some minor revisions must be taken in consideration and some issues need to be addressed:

  • Include further details on the abstract. For instance, define what is the nonlethal concentration and describe what developmental and behavioural parameters were evaluated.
  • Although the second paragraph is well written, there is information missing. For instance, what are the levels of this BPA analogue in water sources? There are some references in the literature in which this compound has been described in wastewaters and surface waters. The inclusion of these concentrations is necessary to frame the objective of the work.
  • Line 52 and 55, what is the “maximum nonlethal concentration”?
  • Line 69, how was BPAP extracted from media? Further details on the HPLC method are required.
  • Change Fig S1 to the actual measured concentrations rather than presenting a ratio.
  • Line 81, review the temperature used for zebrafish which, usually, is around 28 ºC.
  • Line 83, which abnormalities are observed in early fertilized embryos?
  • Line 86, what is the concentration of DMSO in the SC? Also, why these concentrations of BPAP were selected to be studied? Are they environmental relevant? Why not include a control group only exposed to E3? The justification given in line 200 should be moved to this section.
  • Line 88, be more specific. Rather than referring embryos of <4 hpf, please include the exact time at which the exposure begun.
  • Line 95, it is referred that 20 larvae were used for transcriptomic analysis but in the subtopic 2.4, 25 larvae are referred. Review.
  • Line 105, what is the composition of the sample diluent buffer?
  • Include the wavelengths used to measure hormone and protein levels.
  • Line 140, this subtopic should be moved to before TH measurement as these parameters were recorded every 24h until 120 hpf while the remaining were evaluated at the end of the exposure.
  • Line 144, how was the body length measured? Were the larvae immobilized? Further details are required.
  • Line 157, remove “or” as the test is called Dunnett’s T3.
  • The results section should match the same order of the methods, taking in consideration the change suggested in the previous point.
  • Include the p-values for the comparisons made. Also, rather than only comparing to the control group, multiple comparison among the test groups should be performed and included in the text.
  • Line 174, data should be described as mean and SD not SEM. Also, for non-parametric data, results should be described as median and ranges.
  • Figure 2 could be changed to a table as no differences are observed.
  • Line 203, how was the time to hatch measured? Usually the hatching rate is calculated not the time to hatch.
  • Review the x-axis of the Figure 4. SC is not a concentration.
  • Review the body length measured. Usually, animals of 120 hpf have a body length of more than 3 mm not around 2 mm. ZFIN refers a 4.2 mm total body length for 6 days larvae.
  • Line 243, has this hypothesis for the reduction of T4 levels been described in the literature before?
  • Line 271, there are reports of upregulation of dio2 as a possible compensatory response to the decreased T4 levels which was not observed in the current study. Is there any reason for that?
  • In addition, changes in T4 levels are usually associated to growth retardation and malformations of zebrafish larvae which was not observed. Is there any reason to not observed such malformations?
  • Remove citations for figures and tables from the discussion.

Author Response

This study is focused on the effects of bisphenol AP (BPAP), a native bisphenol analogue, in zebrafish development by evaluating developmental, behavioural, biochemical, and genetic parameters after exposure to different BPAP concentrations. The developmental and behavioural outcomes were not affected by the exposure and no changes were observed in the transcriptional profile of different genes although slight changes. The main effect observed was the increased T4 hormone levels which translated as a weak or negligible TH disruption of this compound in zebrafish.

 Overall, the manuscript present novel and significant scientific data for the risk assessment of this alternative to bisphenol and, despite being a well-designed work presenting an interesting and cutting-edge approach, some minor revisions must be taken in consideration and some issues need to be addressed:

Comment #1. Include further details on the abstract. For instance, define what is the nonlethal concentration and describe what developmental and behavioural parameters were evaluated.

Response/Action:

Unfortunately, due to the abstract word limit(maximum 200 words), we could not include those detailed information you pointed out. The manuscript is still under that restriction, and no more rooms for further details.

Nonleathal concentration was decided based on the stastical significance (p < 0.05). The mortality, hatching, malformation and growth were parameters for development. The distance moved and moving duration were the behavioral parameters. We have vividly described those information in the Materials and Methods section.

Comment #2. Although the second paragraph is well written, there is information missing. For instance, what are the levels of this BPA analogue in water sources? There are some references in the literature in which this compound has been described in wastewaters and surface waters. The inclusion of these concentrations is necessary to frame the objective of the work.

Response/Action:

Following your suggestion, we added environmental concnetraions of BPAP in surface water and waste water treatment plants (WWTPs).

After the revision (Line 54-59, page 2)

In addition, BPAP was detected up to 56 ng/L and 1.2 ng/g dw in surface water from Luomo lake and sediment samples from Taihu lake, respectively [13]. In the Korean domestic waste water treatment plants, the measured concentration of BPAP was up to 16.0 ng/g dw [14]. The environmental levels and detection frequencies of BPAP were not higher than BPA and major alternatives of BPA, such as BPF and BPS [13,14].

Comment #3. Line 52 and 55, what is the “maximum nonlethal concentration”?

Response/Action:

This “maximum nonlethal concentration” means that maximum concentration which has no significant effect on lethality and was calculated from preliminary-range finding test. However, this term may not be appropriate for cell viablity. Thus, we revised it “maximum non-cytotoxic dose”

Before the revision (Line 52-56, page 2)

These previous studies revealed that BPAP at levels up to the maximum nonlethal concentration did not affect gene transcription in rat thyroid follicular cells (FRTL-5) [13], and the proliferation of rat pituitary cells (GH3) was not changed under the T3 cotreatment conditions [14]. In contrast, under conditions without T3, treatment with the maximum nonlethal concentration of BPAP increased GH3 cell proliferation and decreased gene transcription (trα, trβ, and dio2) [13,14].

After the revision (Line 61-66, page 2)

These previous studies revealed that BPAP at levels up to the maximum non-cytotoxic dose did not affect gene transcription in rat thyroid follicular cells (FRTL-5) [15], and the proliferation of GH3 cells was not changed under the T3 cotreatment conditions [16]. In contrast, under conditions without T3, treatment with the maximum non-cytotoxic dose of BPAP increased GH3 cell proliferation and decreased gene transcription (trα, trβ, and dio2) [15,16].

Comment #4. Line 69, how was BPAP extracted from media? Further details on the HPLC method are required.

Response/Action:

To extract the BPAP, vortexing and centrifuging were used. Following your suggestion, we added more details for chemical analysis during the revision.

Before the revision (Line 68-78, page 2)

BPAP (CAS RN. 1571-75-1) was purchased from Sigma Aldrich (St. Louis, MO, USA) (Table S1) and was dissolved in dimethyl sulfoxide (DMSO) at 0.1% v/v. The exposure media were collected before and after the media was changed, and the actual concentrations of BPAP in the exposure media were measured by high-performance liquid chromatography (HPLC; Agilent 1260 Infinity, Agilent Technologies, Palo Alto, CA, USA) coupled with an autosampler and diode array detector. The separation was performed using a Waters SunFireTM C18 column (100 mm x 4.6 mm, 5 μm) with an isocratic mobile phase consisting of 30% water and 70% methanol. The flow rate was set to 1 mL/min. The autosampler and column oven temperatures were maintained at 4 °C and 50 °C, respectively. The detection wavelength was set to 275 nm, and the final injection volume of all the samples was 25 μL. The measured concentrations are shown in Fig. S1 and were used to present the results obtained in our study.

After the revision (Line 83-97, page 2-3)

BPAP (CAS RN. 1571-75-1) was purchased from Sigma Aldrich (St. Louis, MO, USA) (Table S1) and was dissolved in dimethyl sulfoxide (DMSO) at 0.1% v/v. The exposure media were collected before and after the media was changed. It was vortexed weakly and separated by centrifuging for 5 min at 12000 rpm. Then, the samples were filtered through 0.22 μm filters for quantification.

The actual concentrations of BPAP in the exposure media were measured by high-performance liquid chromatography (HPLC; Agilent 1260 Infinity, Agilent Technologies, Palo Alto, CA, USA) coupled with an autosampler and diode array detector. The separation was performed using a Waters SunFireTM C18 column (100 mm x 4.6 mm, 5 μm) with an isocratic mobile phase consisting of 30% water and 70% methanol. The flow rate was set to 1 mL/min. The autosampler and column oven temperatures were maintained at 4 °C and 50 °C, respectively. The detection wavelength was set to 275 nm, and the final injection volume of all the samples was 25 μL. The calibration curve for quantifying BPAP was linear, in the range of 10-3000 ng/mL, and the limit of quantification (LOQ) was determined as LOQ = 10 x (SD / Slope), where SD is the standard deviation of the response, and slope is slope of the calibration curve. The measured concentrations are shown in Fig. S1 and were used to present the results obtained in our study.

Comment #5. Change Fig S1 to the actual measured concentrations rather than presenting a ratio.

Response/Action:

Revised following your suggestion.

Before the revision (Figure S1)

Figure S1. Ratio (%) of measured concentrations of BPAP compared with the nominal concentrations. The results are expressed as percentages (%). N.D.: BPAP was not detected in the SC group.

After the revision (Figure S1)

Figure S1. Measured concentrations of BPAP. The LOQ for BPAP in calibration curve was 8.5 ng/mL in this study. N.D.: BPAP was not detected in the SC group.

Comment #6. Line 81, review the temperature used for zebrafish which, usually, is around 28 ºC.

Response/Action:

Yes, you are right. That was our typo. We raised zebrafish and conducted exposure experiment under 28 ± 1 °C (Line 100).

Comment #7. Line 83, which abnormalities are observed in early fertilized embryos?

Response/Action:

During the early developmental stage, the eggs coagulated or contaminated were excluded since those ones cannot develop normally. We revised the sentence to make it clear.

Before the revision (Line 82-83, page 2)

The embryos were collected and inspected under a stereomicroscope to exclude those with abnormalities.

After the revision (Line 101-102, page 3)

The embryos were collected and inspected under a stereomicroscope to exclude those coagulated or contaminated.

Comment #8. Line 86, what is the concentration of DMSO in the SC? Also, why these concentrations of BPAP were selected to be studied? Are they environmental relevant? Why not include a control group only exposed to E3? The justification given in line 200 should be moved to this section.

Response/Action:

DMSO concentration used in this study was 0.1 % v/v and it was presented in Line 69 of our submitted manuscript. We determined exposure concentration of BPAP based on our preliminary test. In our preliminary test, BPAP exposure caused significant lethality at 398.9 µg/L of BPAP but not at 105.9 µg/L of BPAP. At 105.9 µg/L of BPAP, survival rate of zebrafish larvae was > 95%. And we added this information in the Methods section following your comment. The exposure levels in our study is not environmental relevant because our goal of this study was to find out thyroid hormone disrupting potential of BPAP. And we used DMSO based solvent control to compare TH potential with other bisphenols, i.e., BPA, BPF, BPS and BPZ (Lee et al., 2019)[1], because the parameters used in the previous study were also compared with DMSO based solvent control group. As far as we know, using 0.1 % of DMSO is definitely acceptable as solvent control group.

After the revision (Line 106-108, page 3)

In our preliminary test, significant lethal effects were observed at 398.9 µg/L of BPAP but not at 105.9 µg/L of BPAP (>95% survival rate was observed.).

Comment #9. Line 88, be more specific. Rather than referring embryos of <4 hpf, please include the exact time at which the exposure begun.

Response/Action:

Generally, the embryos <4 hpf were used for the test. However, we revised it with exact time, i.e., ≤ 3 hpf (Line 109) following your suggestion. The time from fertilization to exposure was within 3 hours. OCED TG236 also suggests ≤ 3 hpf.

Comment #10. Line 95, it is referred that 20 larvae were used for transcriptomic analysis but in the subtopic 2.4, 25 larvae are referred. Review.

Response/Action:

25 zebrafish larvae were allocated as one replicate. We revised our error in Line 116.

Comment #11. Line 105, what is the composition of the sample diluent buffer?

Response/Action:

It was 1 x PBS. We revised it (Line 126).

Comment #12. Include the wavelengths used to measure hormone and protein levels.

Response/Action:

We added those information following your suggestion.

Before the revision (Line 107-113, page 3)

The levels of hormones were quantified using ELISA kits (Cusabio Biotech, Wuhan, China) for T4 (Cat no. CSB-E08489f), T3 (Cat no. CSB-E08488f) and TSHβ (Cat no. CSB-EQ02721FI) with a BioTek Cytation 5 system (BioTek, Winooski, VT, USA) according to the manufacturer’s recommended protocol. The limits of detection for T4, T3, and TSHβ are reportedly 20 ng/mL, 0.5 ng/mL, and 2.5 µIU/mL, respectively. The measurements of T4, T3, and TSHβ were normalized to the protein level (mg/mL) and analyzed using a BCA protein assay kit (Thermo Fisher Scientific, Johannesburg, South Africa).

After the revision (Line 128-136, page 3)

The levels of hormones were quantified using ELISA kits (Cusabio Biotech, Wuhan, China) for T4 (Cat no. CSB-E08489f), T3 (Cat no. CSB-E08488f) and TSH (Cat no. CSB-EQ027261FI) according to the manufacturer’s recommended protocol. The optical density for T4, T3 and TSH was determined using BioTek Cytation 5 system (BioTek, Winooski, VT, USA) set to 450 nm. The limits of detection for T4, T3, and TSHβ are reportedly 20 ng/mL, 0.5 ng/mL, and 2.5 µIU/mL, respectively. The measurements of T4, T3, and TSHβ were normalized to the protein level (mg/mL). Protein levels were analyzed using a BCA protein assay kit (Thermo Fisher Scientific, Johannesburg, South Africa) and BioTek Cytation 5 system (BioTek, Winooski, VT, USA) set to 562 nm.

Comment #13. Line 140, this subtopic should be moved to before TH measurement as these parameters were recorded every 24h until 120 hpf while the remaining were evaluated at the end of the exposure.

Response/Action:

Thank you for the suggestion. You are right. Some of these parameters were checked every 24 h. In this study, however, we focused on the thyroid hormone rather than other effects. Thus, we wanted to put the most relavant results regarding thryoid hormone disruption first. Please understand our decision on the sequence of our result presentation.

Comment #14. Line 144, how was the body length measured? Were the larvae immobilized? Further details are required.

Response/Action:

Following your suggestion, further details for body length measurement were added in the manuscript.

Before the revision (Line 143-144, page 4)

After exposure, the body lengths and eyeball sizes of the zebrafish larvae were also measured (n = 5).

After the revision (Line 166-170, page 4)

After exposure, the body lengths and eyeball sizes of the zebrafish larvae were also measured (n = 5). For the measurement, the larvae were anesthetized by tricaine methane sulfonate (0.004% w/v) and transferred to a glass slide containing methylcellulose. Then, individual images were taken using a Leica M205FA fluorescent microscope mounted with a Leica DFC 7000T camera module (Leica Camera AG, Wetzlar, Germany).  

Comment #15. Line 157, remove “or” as the test is called Dunnett’s T3.

Response/Action:

To make the meaning clear, we revised the sentence.

Before the revision (Line 156-158, page 4)

To identify significant differences between the treatments and the SC, one-way analysis of variance (ANOVA) with Dunnett’s or T3 post hoc analysis was performed using SPSS 12.0 for Windows® (SPSS, Chicago, IL, USA).

After the revision (Line 182-185, page 4)

To identify significant differences between the treatments and the SC, one-way analysis of variance (ANOVA) with Dunnett and Dunnett’s T3 post hoc analysis was performed using SPSS 12.0 for Windows® (SPSS, Chicago, IL, USA) for equal variances and unequal variances, respectively.

Comment #16. The results section should match the same order of the methods, taking in consideration the change suggested in the previous point.

Response/Action:

Thank you for the suggestion. As we responded to your comment #13, we wanted to put more important data first in terms of thyroid hormone disruption. Please, understand our decision.

Comment #17. Include the p-values for the comparisons made. Also, rather than only comparing to the control group, multiple comparison among the test groups should be performed and included in the text.

Response/Action:

Thank you for the suggestion. However, marking asterisks (p < 0.05) without each detailed p-value and comparison with only (solvent) control group are generally acceptable and provide appropriate information. Those are applied in the similar previous studies i.e., Huang et al., 2016[2] and Zhang et al., 2017[3], etc.

Comment #18. Line 174, data should be described as mean and SD not SEM. Also, for non-parametric data, results should be described as median and ranges.

Response/Action:

Following your comment, we presented our data with a median and ranges for non-parametric data (revised Table 1). Gene transcription data for tg and dio3 were only non-parametric data in our study. However, many previous scientific articles also used mean and SEM value. It has been widely used. As far as we know, SEM is still acceptable way since there are no golden rules for that issue yet.

After the revision (Line 30-31, page 2)

Table 1. TH-related gene transcription levels (fold changes) in whole-body homogenates of zebrafish larvae after BPAP exposure ((a) thyroid stimulation, (b) TH synthesis, (c) TH receptors and transport, and (d) TH metabolism).

Conc.

(μg/L)

Thyroid stimulation

TH receptors and transport

crh

tshβ

trα

trβ

ttr

0

1.00 ± 0.21

1.00 ± 0.23

1.00 ± 0.34

1.00 ± 0.08

1.00 ± 0.08

18.2

0.72 ± 0.18

0.79 ± 0.27

1.43 ± 0.37

1.02 ± 0.24

1.00 ± 0.15

43.4

0.89 ± 0.16

0.77 ± 0.26

1.83 ± 0.31

1.21 ± 0.20

1.15 ± 0.26

105.9

1.13 ± 0.10

1.29 ± 0.20

1.57 ± 0.28

1.03 ± 0.25

1.16 ± 0.22

Conc.

(μg/L)

TH synthesis

nkx2.1

hhex

tshr

slc5a5

tga

Pax8

tpo

0

1.00 ± 0.23

1.00 ± 0.15

1.00 ± 0.19

1.00 ± 0.17

1.02

(0.92-1.07)

1.00 ± 0.26

1.00 ± 0.08

18.2

0.79 ± 0.25

0.83 ± 0.16

0.72 ± 0.25

0.60 ± 0.23

0.86

(0.35-1.01)

0.72 ± 0.10

0.76 ± 0.06

43.4

0.86 ± 0.26

0.94 ± 0.13

0.77 ± 0.31

0.55 ± 0.23

1.03

(0.93-2.25)

0.92 ± 0.15

0.98 ± 0.07

105.9

1.26 ± 0.26

1.21 ± 0.27

1.19 ± 0.27

1.04 ± 0.27

1.60

(1.47-2.88)

0.78 ± 0.19

0.81 ± 0.11

Conc.

(μg/L)

TH metabolism

dio1

dio2

dio3a

ugt1ab

0

1.00 ± 0.19

1.00 ± 0.25

0.99

(0.89-1.12)

1.00 ± 0.17

18.2

0.97 ± 0.03

0.67 ± 0.22

0.79

(0.45-1.16)

0.75 ± 0.10

43.4

1.11 ± 0.13

0.93 ± 0.01

0.88

(0.71-0.88)

0.91 ± 0.08

105.9

1.11 ± 0.03

1.13 ± 0.28

1.03

(0.87-2.29)

1.17 ±0.19

The results are shown as the means ± SEMs (N=3). None of the effects were statistically significant compared with the solvent control (0.1% v/v DMSO) (p < 0.05). a: Non-parametric data shown as the median with ranges.

Comment #19. Figure 2 could be changed to a table as no differences are observed.

Response/Action:

Revised following your suggestion. Please refer to your previous comment #18.  

Comment #20. Line 203, how was the time to hatch measured? Usually the hatching rate is calculated not the time to hatch.

Response/Action:

We also checked hatching rate and found all the surviving embryos were successfully hatched (Line 206 of the submitted manuscript). Time to hatch can also be one of the indicator for developmental delay. To check “time to hatch”, we recorded hatchability every day and counted how many fishes were hatched each day. Based on the records, we calculated the “time to hatch” in this study.

Comment #21. Review the x-axis of the Figure 4. SC is not a concentration.

Response/Action:

Yes, you are right. Revised as you suggested in the Figure 1, 3, 4, S1 and S2.

Comment #22. Review the body length measured. Usually, animals of 120 hpf have a body length of more than 3 mm not around 2 mm. ZFIN refers a 4.2 mm total body length for 6 days larvae.

Response/Action:

Following your comment, we checked our data again and we found an error on microscope magnification (X2). We rectify it. Thank you for the check.

Before the revision (Figure 4, page 8)

(b)

After the revision (Figure 3, page 8)

(b)

Comment #23. Line 243, has this hypothesis for the reduction of T4 levels been described in the literature before?

Response/Action:

Following your suggestion, we added related references in Line 274. Zhang et al., 2017 (ref #1), Guo et al., 2013 (ref #32) and Wang et al., 2013 (ref #33) explained the changes of T4 level with TH synthesis and metabolism (degradation).

Comment #24. Line 271, there are reports of upregulation of dio2 as a possible compensatory response to the decreased T4 levels which was not observed in the current study. Is there any reason for that?

Response/Action:

In our study, significant upregulation of dio2 was not shown. In addition, the level of T3 was not changed, too. This might mean that T4 reduction in our study was not only due to dio2 transcription. Thus, the reason for that is not clear yet.

On the other hand, T4 reduction at 105.9 μg/L of BPAP exposure was significant but other parameters were not changed. It might mean that T4 reduction in our study might be quite weak or negligible physiologically despite the statistical outcome.

Comment #25. In addition, changes in T4 levels are usually associated to growth retardation and malformations of zebrafish larvae which was not observed. Is there any reason to not observed such malformations?

Response/Action:

Yes, you are right. T4 level changes can affect growth, development, and even behavior. Thus, we applied those parameters also in our study. But, those parameters were not affected by BPAP exposure. Thus, we can conclude that the thyroid disrupting effects of BPAP was physiologically weak or negligible despite T4 reduction as we mentioned in your comment #24.

Comment #26. Remove citations for figures and tables from the discussion.

Response/Action:

We believe that citations for figures and tables in the discussion can help readers to check results easily and to follow the flow of contents. If it is acceptable, we would like to keep those citations in the discussion session. Thanks for the detailed comment.

Reviewer 2 Report

Lee et al., investigated the effects of a bisphenolic compound BPAP on the thyroid system in zebrafish larvae. The authors have found that BPAP has weak or negligible potency regarding TH disruption. Overall, the manuscript is well-written and easy to follow. However, some questions need to be addressed before publication.
The purpose of the study is not clear. In fact, the rationale and problem statement are not that much strong. There are only two previous studies pointing to BPAP as a weak TH disruptor!
I recommend the authors to provide more information about the adverse effects of other bisphenols such as BPA and BPS on the thyroid system in zebrafish. It is known while these compounds are weak estrogenic chemicals, they have a considerable potential to disrupt THs.
Moreover, I encourage the authors to provide more background regarding the importance of TH in zebrafish development.
Why did the authors choose 4-120 hpf window for exposure! it should be cleared in the MS.
Were the kits used for T3 and T4 measurement specifically designed for zebrafish?
It would be more informative if the authors provide some information regarding environmental levels of BPAP

Author Response

Lee et al., investigated the effects of a bisphenolic compound BPAP on the thyroid system in zebrafish larvae. The authors have found that BPAP has weak or negligible potency regarding TH disruption. Overall, the manuscript is well-written and easy to follow. However, some questions need to be addressed before publication.

Comment #1. The purpose of the study is not clear. In fact, the rationale and problem statement are not that much strong. There are only two previous studies pointing to BPAP as a weak TH disruptor!

Response/Action:

Yes, you are right. There were only two related previous studies as you mentioned. However, this present study started from the facts that many of bisphenol A alternatives, i.e., BPF, BPS have thyroid hormone disrupting potency as well as BPA. We realized that the effects of BPAP on thyroid hormone system was ambiguous based on the two in vitro studies. None of the in vivo studies was found for BPAP. Therefore, we wanted to evaluate thyroid hormone disrupting effects of BPAP by using an in vivo zebrafish model which may provide more comprehensive output (Line 66-69).

Comment #2. I recommend the authors to provide more information about the adverse effects of other bisphenols such as BPA and BPS on the thyroid system in zebrafish. It is known while these compounds are weak estrogenic chemicals, they have a considerable potential to disrupt THs. Moreover, I encourage the authors to provide more background regarding the importance of TH in zebrafish development.
Response/Action:

Following your suggestion, we added more information about the effect of BPAP on TH and the importance of TH systems in zebrafish development.

Before the revision (Line 30-36, page 1)

The endocrine-disrupting effects of bisphenol A (BPA) and its major alternatives, such as bisphenol F (BPF) and bisphenol S (BPS), have recently been reported [1-3]. The effects of these chemicals on the disruption of the thyroid hormone (TH), an integral endocrine system, need to be evaluated because THs play essential roles in many physiological processes, such as development, growth, reproduction and metabolism [3,4]. BPA and bisphenols, such as bisphenol AF (BPAF), BPF, and BPS, induce disruption of the TH regulation system [1-5]. To find alternative chemicals that are relatively safer than BPA with respect to TH disruption, various candidates need to be evaluated.

After the revision (Line 30-41, page 1)

The endocrine-disrupting effects of bisphenol A (BPA) and its major alternatives, such as bisphenol F (BPF) and bisphenol S (BPS), have recently been reported [1-3]. As one of the integral endocrine system, thyroid hormone (TH) disrupting effects of those chemicals need to be evaluated because THs play essential roles in many physiological processes, such as development, growth, reproduction and metabolism [3,4]. BPA and bisphenols, such as bisphenol AF (BPAF), BPF, and BPS, induce disruption of the TH regulation system [1-5]. In our previous study, BPA, BPF and BPS induced the level of T3 or T4 and changed related gene transcriptions in the zebrafish embryo/larvae model [3]. BPA, BPF and BPS also induced TH-dependent rat pituitary cells (GH3) proliferation and biphasic responses of gene transcription in Pelophylax nigromaculatus tadpoles [2]. However, T4 level reduction was also shown by BPF and BPS [1,4,5]. Despite conflicting results for bisphenols regarding THs, TH disrupting effects of those bisphenols are evident. To find alternative chemicals that are relatively safer than BPA with respect to TH disruption, various candidates need to be evaluated.

Before the revision (Line 56-59, page 2)

Because the TH-disrupting effects of BPAP remain unclear, an evaluation of TH disruption using more comprehensive in vivo models, e.g., zebrafish, is needed because these models have a complete TH regulation system, such as hypothalamus-pituitary-thyroid (H-P-T) feedback.

After the revision (Line 66-74, page 2)

Because the TH-disrupting effects of BPAP remain unclear, an evaluation of TH disruption using more comprehensive in vivo models, e.g., zebrafish, is needed because these models have a complete TH regulation system, such as hypothalamus-pituitary-thyroid (H-P-T) feedback. As a non-animal alternative model for chemical screening, zebrafish embryos can be used until the zebrafish starts to feed independently, which is at 5 d post fertilization (dpf) [17]. Despite its morphological differences in mature thyroid gland compared to higher vertebrates, the early steps in thyroid development of zebrafish show a significant resemblance [18,19]. From 70 to 80 h post fertilization (hpf), T4 synthesis in zebrafish embryo becomes detectable [19,20].

Comment #3. Why did the authors choose 4-120 hpf window for exposure! it should be cleared in the MS.
Response/Action:

Accoding to the Directive 2010/63/EU, zebrafish embryos can be used as a non-animal alternative model for chemical screening until the zebrafish starts to feed independently, which is at 5 days post fertilization (dpf). In addition, thyroid development and thyroid hormone synthesis were detectable between 70 to 80 hours post fertilization (hpf). Thus, the exposure window from 4 to 120 hpf is widely accepted in the previous studies for evaluating thyroid hormone disruption. We added background information for this in the Introduction section.

After the revision (Line 69-74, page 2)

As a non-animal alternative model for chemical screening, zebrafish embryos can be used until the zebrafish starts to feed independently, which is at 5 d post fertilization (dpf) [17]. Despite its morphological differences in mature thyroid gland compared to higher vertebrates, the early steps in thyroid development of zebrafish show a significant resemblance [18,19]. From 70 to 80 h post fertilization (hpf), T4 synthesis in zebrafish embryo becomes detectable [19,20].

Comment #4. Were the kits used for T3 and T4 measurement specifically designed for zebrafish?
Response/Action:

All the ELISA kits used in this study were designed for fish species. Previous studies using the zebafish model, e.g., BPF or BPS thyroid hormone disruption studies, also used exactly the same kits. In our internal test, the linearity among diluted zebrafish samples were reliable (data not shown).  

Comment #5. It would be more informative if the authors provide some information regarding environmental levels of BPAP

Response/Action:

We added environmental levels of BPAP as advised.

After the revision (Line 54-59, page 2)

In addition, BPAP was detected up to 56 ng/L and 1.2 ng/g dw in surface water from Luomo lake and sediment samples from Taihu lake, respectively [13]. In the Korean domestic waste water treatment plants, the measured concentration of BPAP was up to 16.0 ng/g dw [14]. The environmental levels and detection frequencies of BPAP were not higher than BPA and major alternatives of BPA, such as BPF and BPS [13,14].

Reviewer 3 Report

a) Introduction section: A brief description of bisphenols uses, in general and in particular, should be given. Most of them ar used as plasticizers, but some of them are used, for example, (only) as flame retardants. All bisphenols currently used at industrial level should be listed/mentioned. 

b) Subsection 2.1: The limit of detection or limit of quantification concerning the HPLC-DAD analytical method for the determination of BPAP should be mentioned.

c)  Line 85: the chemical formula for calcium chloride is CaCl2, not CaCl; CaCl2, CaCl2*6H2O or CaCl2*2H2O was used ? - to be checked; the concentration of the culture medium can be different, depending on the type of salt used;

d) Line 85: to be checked; what kind of magnesium sulfate was used ? anhydrous (MgSO4), monohydrate (MgSO4*H2O) or heptahydrate (MgSO4*7H2O) ? the concentration of the medium can be different, depending on the type of salt used;

Author Response

Comment #1. Introduction section: A brief description of bisphenols uses, in general and in particular, should be given. Most of them are used as plasticizers, but some of them are used, for example, (only) as flame retardants. All bisphenols currently used at industrial level should be listed/mentioned. 

Response/Action:

Thank you for the constructive comment. In the introduction section, the information which you pointed out was already described for BPAP. During this revision, we added more information for BPAP, such as levels in the environment. However, we did not cover the information for other bisphenols. It is due to that other previous papers (e.g., ref #8 and #9) already covered those information and it might make the purpose of this study unclear. We believe that it would be better to focuse on our purpose, i.e., thyroid hormone disrupting potential of BPAP, more in the Introduction section.

Comment #2. Subsection 2.1: The limit of detection or limit of quantification concerning the HPLC-DAD analytical method for the determination of BPAP should be mentioned.

Response/Action:

Following your suggestion, we added details for LOQ in methods and Fig S1.

Before the revision (Line 76-78, page 2)

The detection wavelength was set to 275 nm, and the final injection volume of all the samples was 25 μL. The measured concentrations are shown in Fig. S1 and were used to present the results obtained in our study.

After the revision (Line 92-97, page 2-3)

The detection wavelength was set to 275 nm, and the final injection volume of all the samples was 25 μL. The calibration curve for quantifying BPAP was linear in the range of 10-3000 ng/mL, and the limit of quantification (LOQ) was determined as LOQ = 10 x (SD / Slope), where SD is the standard deviation of the response, and slope is slope of the calibration curve. The measured concentrations are shown in Fig. S1 and were used to present the results obtained in our study.

Before the revision (Figure S1 Title)

Figure S1. Ratio (%) of measured concentrations of BPAP compared with the nominal concentrations. The results are expressed as percentages (%). N.D.: BPAP was not detected in the SC group.

After the revision (Figure S1 Title)

Figure S1. Measured concentrations of BPAP. The LOQ for BPAP in calibration curve was 8.5 ng/mL in this study. N.D.: BPAP was not detected in the SC group.

Comment #3.  Line 85: the chemical formula for calcium chloride is CaCl2, not CaCl; CaCl2, CaCl2*6H2O or CaCl2*2H2O was used? - to be checked; the concentration of the culture medium can be different, depending on the type of salt used;

Response/Action:

“CaCl” was our typo. And we used 0.044 g of CaCl2•2H2O for E3 media. Thank you for the detailed check.

Comment #4. Line 85: to be checked; what kind of magnesium sulfate was used ? anhydrous (MgSO4), monohydrate (MgSO4*H2O) or heptahydrate (MgSO4*7H2O) ? the concentration of the medium can be different, depending on the type of salt used;

Response/Action:

For MgSO4, we used the heptahydrate form (MgSO4•7H2O). We revised it. Thank you for the detailed check.